# EZH2: An Accomplice of Gastric Cancer

**DOI:** 10.3390/cancers15020425

**Published:** 2023-01-09

**Authors:** Wuhan Yu, Ning Liu, Xiaogang Song, Lang Chen, Mancai Wang, Guohui Xiao, Tengfei Li, Zheyuan Wang, Youcheng Zhang

**Affiliations:** 1The Second Hospital of Lanzhou University, Lanzhou 730030, China; 2The Second General Surgery Department, The Second Hospital of Lanzhou University, Lanzhou 730030, China; 3The Neurology Department, The Second Hospital of Lanzhou University, Lanzhou 730030, China; 4The Cardiology Department, The Second Hospital of Lanzhou University, Lanzhou 730030, China

**Keywords:** gastric cancer, enhancer of zeste homolog 2, H3K27, epigenetics

## Abstract

**Simple Summary:**

Enhancer of zeste homolog 2 (EZH2) modifies the trimethylation of Lys-27 of histone 3, affecting downstream target genes’ expression. It was reported that EZH2 is highly expressed in gastric cancer and may be a potential prognostic molecule and promising therapeutic target. We aim to present the value of EZH2 research in gastric cancer by focusing on the crucial events of EZH2 involvement in gastric cancer progression. Therefore, in this review, we present the two main functions of EZH2: histone methylation modification and DNA methylation by EZH2; the molecular mechanism of the action of EZH2 in regulating target genes; a detailed description of the mechanism of EZH2 in gastric cancer-related events. Finally, progress in the development of EZH2 inhibitors is summarized. This review article provides researchers studying the epigenetics of gastric cancer with research ideas to find new targets for studying gastric cancer pathogenesis.

**Abstract:**

Gastric cancer is the fifth most common cancer and the third leading cause of cancer deaths worldwide. Understanding the factors influencing the therapeutic effects in gastric cancer patients and the molecular mechanism behind gastric cancer is still facing challenges. In addition to genetic alterations and environmental factors, it has been demonstrated that epigenetic mechanisms can also induce the occurrence and progression of gastric cancer. Enhancer of zeste homolog 2 (EZH2) is the catalytic subunit of the polycomb repressor complex 2 (PRC2), which trimethylates histone 3 at Lys-27 and regulates the expression of downstream target genes through epigenetic mechanisms. It has been found that EZH2 is overexpressed in the stomach, which promotes the progression of gastric cancer through multiple pathways. In addition, targeted inhibition of EZH2 expression can effectively delay the progression of gastric cancer and improve its resistance to chemotherapeutic agents. Given the many effects of EZH2 in gastric cancer, there are no studies to comprehensively describe this mechanism. Therefore, in this review, we first introduce EZH2 and clarify the mechanisms of abnormal expression of EZH2 in cancer. Secondly, we summarize the role of EZH2 in gastric cancer, which includes the association of the EZH2 gene with genetic susceptibility to GC, the correlation of the EZH2 gene with gastric carcinogenesis and invasive metastasis, the resistance to chemotherapeutic drugs of gastric cancer mediated by EZH2 and the high expression of EZH2 leading to poor prognosis of gastric cancer patients. Finally, we also clarify some of the current statuses of drug development regarding targeted inhibition of EZH2/PRC2 activity.

## 1. Introduction

Gastric cancer (GC) is the fifth most diagnosed malignancy worldwide, with more than 1 million new cases annually [1]. There is a lack of methods to diagnose GC early, so many patients are diagnosed at a later stage, which leads to a high mortality rate for GC patients [1]. It was reported that more than 784,000 GC patients died worldwide in 2018, making it the third most common cause of death among oncologic diseases [2]. The complicated pathogenesis, late diagnosis and lack of effective treatment for GC lead to the poor prognosis of patients. To fundamentally prevent and treat GC, it is highly significant to understand the pathogenesis of GC [3]. In addition to genetic changes and environmental factors, it has been proven that epigenetic inheritance guides the occurrence and development of cancer and is a hallmark of gastric malignancies [4]. It was known that the polycomb group (PcG) was one of the most important epigenetic regulators, which influences the expression of many genes involved in the development of the body [5]. As a core member of the PcG family, EZH2 plays a vital role in cell proliferation, differentiation and tumor formation through H3K27me3-mediated downstream gene silencing. The expression of H3K27me3 in GC tissues is significantly increased, and it is the most common type of histone methylation modification in GC studies, which is closely related to the pathogenesis of GC and the prognosis of patients [4,6]. In conclusion, EZH2 plays a role in the pathogenesis of GC through H3K27me3. For these reasons, EZH2 can be considered an exciting target for developing targeted therapies for GC. Therefore, this article focuses on the relationship between EZH2 and the occurrence, metastasis, and drug resistance of GC and further explores the mechanisms of development of GC from the epigenetic factors.

## 2. Overview of EZH2

PcG proteins are a group of transcriptional repressors that regulate target genes through chromatin modification and can induce tumor development. These proteins chemically and functionally represent the core proteins of the polycomb repressive complexes (PRCs) [5]. PRCs are enzyme complexes that modify lysine residues on histones [7,8]. There are two major PRCs in mammals: Polycomb repressive complex 1 (PCR1) and polycomb repressive complex 2 (PCR2). PRC1 consists of ring finger protein 1 (RING 1) (RING1A or RING1B) and PcG ring finger protein (PCGF1-6) that monoubiquitinates lysine 119 on histone H2A (H2AK119ub1) [9]. PRC2 complexes are histone methyltransferases (HMTases) that are dependent on S-adenosyl-L-methionine (SAM) and contain four major core subunits: EZH2/1, suppressor of zeste 12 (SUZ12), embryonic ectoderm development 1-4 (EED1-4) and RBAP46/48 [9,10]. It catalyzes the mono-methylation, di-methylation and tri-methylation of lysine 27 on histone H3 (H3K27me1, H3K27me2 and H3K27me3 [11]. PRC2 is further divided into two different subclasses: PRC2.1 and PRC2.2. In addition to the four major core structures, the former includes Jumonji AT-rich interactive domain 2 (JARID2) and adipocyte enhancer binding protein 2 (AEBP2), the latter of which includes PCL1-3 and c17orf96/c10orf12 [11] (Figure 1). At present, the accessory proteins have been shown to regulate PRC2 activity and play a role in cells by localizing PRC2 to chromatin. However, the exact function of these proteins is unknown. 

EZH2 is a critical functional member of the PRCs family, which is located in chromosome 7q35 and consists of 20 exons containing 746 amino acid residues [12] (Figure 2C). It has five structural domains, including the EED-interacting structural domain (EID), structural domain I, structural domain II (SANT2L), a cysteine-rich structural domain (CXC structural domain), and a three-chested structural domain (SET structural domain) [12,13] (Figure 2D). Its most critical function is to inhibit gene expression by promoting histone methylation and DNA methylation in the nucleus.

## 3. Histone Modification of EZH2

Histones are an essential part of the nucleosome, the basic structure of chromosomes. The modification of histones can change the loose or agglutination state of chromatin which has a regulatory effect on gene expression similar to that of the DNA genetic code [14]. It was clarified that H3K27me3 is considered a key epigenetic event that can make the chromosome structure denser to inhibit the expression of target genes [15]. EZH2, a histone methyltransferase in the human genome, catalyzes the lysine trimethylation of histone 3 at position 27 (H3K27me3), which leads to the silencing of its target genes involved in cell proliferation, cell differentiation, and cancer development [16,17]. However, it is noteworthy that EZH2 alone cannot exert its methyltransferase biological activity. EZH2 must be combined with at least two non-catalytic partners, SUZ12 and EED, to obtain a strong histone methyltransferase activity [18,19,20]. In addition, EZH2 typically forms PRC2 complexes to perform histone-modifying processes, which is the classical mode of action of EZH2 and will be explicitly described below. EZH2 is overactive in cancer cells through functionally acquired mutations and overexpression. In a study on prostate cancer, elevated levels of EZH2 and H3K27me3 were associated with poor prognosis in metastatic prostate cancer. However, the deletion of EZH2 inhibited the growth of prostate cancer cells [21]. The overexpression of EZH2 and elevation of H3K27me3 in solid cancers, including breast, gastric, endometrial, ovarian, melanoma, bladder, kidney, colorectal, and lung cancers, as well as hematological malignancies such as T-cell and B-cell lymphomas [22,23]. In conclusion, the histone modification of EZH2 is closely related to tumorigenesis. The overexpression of EZH2 leads to an increase in H3K27me3, which inhibits tumor suppressor genes including p16 and E-cadherin, and drives cellular differentiation [24] (Table 1).

## 4. DNA Methylation of EZH2

DNA methylation is another significant mechanism of epigenetic regulation. DNA hypermethylation at promoter site CpG islands is thought to promote tumorigenesis through transcriptional silencing of tumor suppressor genes [28]. There are some DNA methyltransferases (DNMTs), including DNMT1, DNMT3A, and DNMT3B, which assist in the pattern of DNA methylation. Some studies reported that these genes play a crucial role in various human cancers [26]. Vire et al. found that EZH2 interacts directly with DNMTs and recruits DNMT to the promoter regions of target genes in cancer cells, affecting DNA methylation status, suggesting that EZH2 is involved in DNA methylation [17]. (Table 2). For these reasons, EZH2 is not only essential in histone modification, but also plays a vital role in DNA methylation as EZH2 is thought to be a recruitment platform for DNMT. In fact, histone methylation can help to guide DNA methylation patterns. In addition, DNA methylation might serve as a template for some histone modifications after DNA replication [29]. In general, this suggests that histone modifications and DNA methylation synergistically regulate epigenetic states and promote tumor development.

## 5. Mechanism of Abnormal Expression of EZH2 in Cancer

Epigenetic abnormalities are key factors in the development and progression of cancer [35]. It has been widely recognized that EZH2 can be regulated in different kinds of human cancers at both the transcriptional and translational levels [36]. Next, we will introduce the mechanism of EZH2 in cancer in different ways (Figure 3).

## 6. Regulation of EZH2 in Transcription Levels

### 6.1. Transcription Inhibition Dependent on PcG Family

EZH2 acts primarily as a histone methyltransferase through its SET domain which can inhibit or co-activate transcription in a PRC2-dependent or independent manner. PcG protein, an important epigenetic regulator, is considered a transcriptional repressor and a key regulator of cell fate in cancer development. They include two complexes, PRC1 and PRC2 [37]. It has been reported that EZH2 can be used as a catalytic subunit of PRC2, the complex that participates in the transcriptional inhibition of multiple target genes, including the inactivation of more than 200 tumor suppressor genes [12]. In the process of gene silencing, EZH2, SUZ12, EED, retinoblastoma suppressor associated protein 46/48 (RbAp46/48), and AEBP2 first polymerize to form a PRC2 complex in the nucleus. Subsequently, the catalytic domain SETB (su(var)3-9, enhancer of zeste, trithorax) in EZH2 will catalyze trimethylation of lysine at position 27 of histone H3 (H3K27me3) that binds to the target gene. In addition, the subunit homologue of PCR1, that has the same function as PRC2, recognizes and binds to the methylation site of histone H3. Finally, RING 1, the catalytic subunit of PRC1, further catalyzes lysine monobiquitination at position 119 of histone H2A and inhibits the transcriptional elongation reaction that is dependent on RNA polymerase, thus inhibiting the transcription of target genes [26]. In recent years, with the completion of whole genome sequencing and chromatin immunoprecipitation sequencing, scientists have found that some genes can still be recognized by PRC2 and inhibit transcription even those that lack H2AK119ub1 modification sites [38]. In addition, PRC1 can also independently polymerize to the target gene and mediate H2AK119ub1 generation without the involvement of PRC2 and the methylation of H3 histone [39]. Meanwhile, it was found that the EZH2 target gene labeled with H3K27me3 by PRC2 in normal cells has a corresponding relationship with the abnormal hypermethylation gene in cancer cells. Therefore, it is speculated that under the condition of the existence of carcinogenic inducements, the abnormal increase of EZH2 expression will change the target gene of H3K27me3 marker mediated by PRC2 into a higher methylation state during normal development, thus changing the activity of the target gene and causing the transformation of normal cells into cancer cells [40].

### 6.2. Transcription Inhibition Independent of the PcG Family

There is another non-classical EZH2 regulatory mechanism in cells which is a PRC2-independent way to directly combine with other factors to form transcription complexes to activate the transcription of downstream target genes. It was reported that cycle-related kinases activate the EZH2/NF-κB signaling pathway. Subsequently, EZH2 interacts with NF-κB to form the EZH2–p-p65-Ser536 complex, which binds to the IL-6 promoter and leads to the accumulation of immunosuppressive myeloid-derived suppressor cells (MDSCs), providing a tumor immunosuppressive microenvironment and inducing drug resistance [41]. Moreover, EZH2, along with the potential epigenetic regulator lysine-specific demethylase 2B (KDM2B), dampens colorectal cancer migration, invasion, and maintenance of cancer stem cells via the PI3K/AKT pathway [42]. Kim also found that Akt-mediated serine phosphorylation of EZH2 at position 21 occurs in glioma stem cells, allowing EZH2 to interact with the transcriptional activation factor STAT3 to induce lysine methylation at position 180 [43]. It follows that EZH2 directly interacts physically with several transcription factors in tumor cells and exerts histone methyltransferase activity independently of the PcG family.

It has been reported that RNA may also be involved in the independent regulation of EZH2 transcription. There is negative regulation of the expression of EZH2 by microRNA (miRNA). In tumor cells, down-regulation of miRNA was significantly correlated with EZH2 over-expression because of direct inhibition of the transcription and translation process of EZH2 by miRNA [44]. Moreover, EZH2 can bind to the STAT3 signaling pathway via a novel circular RNA named circ-LRIG3, resulting in methylation and phosphorylation, and promoting the development and progression of hepatocellular carcinoma [45]. In addition, the functions of EZH2 that switches from histone methyltransferase to non-histone methyltransferase rely on long non-coding RNA (lncRNA) p21. Sun reported that STAT3/lncRNA HOTAIR interacts with pEZH2-Ser21 to regulate the growth of head and neck squamous cell carcinoma, thereby improving the anti-tumor efficacy of cisplatin and cetuximab therapy [46,47]. It can be seen that EZH2 plays different roles in regulating the transcription of target genes by interacting with different transcription factors.

## 7. Regulation of EZH2 Translation and Post-Translational Modifications

It has been reported that the two main mechanisms responsible for protein diversity, mRNA and PTMs, lead to the number of proteins far exceeding the estimated DNA coding capabilities [48]. At present, increasing evidence supports that EZH2 can also be regulated by post-translational modifications (PTMs) in the development of cancer. Several studies have confirmed the importance of PTMs in the regulation of tumors by EZH2, particularly those involved in phosphorylation, acetylation, ubiquitination, sumoylation, and O-GlcNAcylation [37]. The scientists found that phosphorylation of T416 mediated by cyclin-dependent kinase 2 promoted EZH2 recruitment at target gene promoters, and the methylation of K307 mediated by SMYD2 enhanced the stability of EZH2 [49,50]. Glycosylation of O-GlcNAc occurs at multiple sites in the EZH2 molecule, such as S73, S84, S87, T313, and S729, which can regulate the level of free EZH2 [51]. In addition, ubiquitination is also vital for EZH2 regulation. For example, the degradation of K63 ubiquitination mediated by tumor necrosis factor receptor-associated factor 6 can reduce EZH2 levels [52]. Collectively, studying the regulation of EZH2 by different types of PTMs and their regulation in carcinogenesis, as well as elucidating its intrinsic molecular mechanism, will open up a potential and promising approach for tumor therapy.

## 8. Role of EZH2 in GC

The occurrence of GC is a multi-stage evolutionary process in which gene research is crucial for the diagnosis and treatment of GCs (Table 3). EZH2 alters the cellular memory system and regulates transcription by targeting silencing mechanisms, which is closely related to the occurrence and development of GCs. Next, we mainly introduce the expression and prognostic effects of EZH2 in GC, and discuss the possible mechanism of EZH2 guiding the progression of GC from aspects of metastasis metabolism and drug resistance.

## 9. Association of the EZH2 Gene with Genetic Susceptibility to GC

In recent years, the relationship between EZH2 single-nucleotide polymorphisms (SNPs) and tumor genetic susceptibility has attracted the attention of many scholars. Variants of SNPs in EZH2 can affect the function of EZH2 and its downstream targets by altering EZH2 transcription and H3K27 trimethylation. Thus SNPs associated with poorer prognosis would be predictors of higher EZH2 expression [58]. Joan found that the frequency of the rs2302427 (D185H) allele of the EZH2 gene was 3.7% and 5.2% in the case and control groups, respectively, and that the EZH2 heterozygous genotype significantly reduced the risk of prostate cancer (OR = 0. 63, P = 0.0085) [59]. Crea, F. et al. showed that the rs3757441 genotype of the EZH2 gene was associated with progression-free survival time and overall survival time in colon cancer. More importantly, some reports focused on the involvement of EZH2 SNPs in GC susceptibility. Zhou et al. investigated the relationship between SNP variants in the EZH2 gene and genetic susceptibility to GC using a case-control study [60]. Their study confirmed that EZH2 variants were significantly associated with GC risk, which provides a new perspective on the susceptibility factors of EZH2 gene variants in gastric carcinogenesis. Their study genotyped EZH2 in 311 cases of GC and 425 cases of the Chinese Han population and found 5 SNPs (rs12670401, rs6464926, rs2072407, rs734005, rs734004) in the EZH2 gene that were significantly associated with the risk of GC development. Among them, the rs12670401 genotype CC and rs6464926 TT can increase the risk of GC. rs12670401 is located at the intron region of the EID that binds with EED, and rs6464926 is located in the D2 intron region of the SUZ12-binding site. Therefore, these two SNP loci polymorphisms may indirectly affect the binding efficiency of EZH2 to EED and SUZ12, thus affecting the formation of PRC2 and further preventing it from playing the role of a histone methyltransferase. The other three loci, rs734004 genotype CG, rs734005 genotype TC, and rs2072407 genotype TC, all reduced the risk of gastric carcinogenesis. In another study, and Lee genotyped 23 tag SNPs of EZH2 in 2349 Korean participants. The SNP genotypes of 1100 patients with GC and 1249 healthy controls were compared to conduct a statistical test for their GC risk correlation and epistasis. The results showed that EZH2 SNPs were associated with susceptibility to GC, the depth of primary tumor invasion, and lymph node metastasis [61], which was consistent with the findings of Sun et al. [62]. In conclusion, gene polymorphisms in EZH2 play a crucial role in the occurrence and development of gastric cancer. However, the current research on the relationship between the gene polymorphisms in EZH2 and gastric cancer is not comprehensive enough, and the research area of relevant experiments is also limited and needs further research.

## 10. Correlation of EZH2 Gene with Gastric Carcinogenesis and Invasive Metastasis

The correlation between the EZH2 gene and cell invasion and metastasis of cancer is a research hotspot. Some studies have shown that the mutation or overexpression of the EZH2 gene is often directly related to the progression of malignant tumors. In recent years, the increased expression of EZH2 in tissues of GC is significantly associated with gastric carcinogenesis, progression, invasion, and metastasis. 

One of the key events is that EZH2 promotes epithelial–mesenchymal transition (EMT) events. EMT is a phenomenon that leads epithelial cells to gradually acquire a mesenchymal phenotype. During this process, the expression of specific epithelial proteins called “epithelial markers” is deficient, such as E-cadherin (encoded by the CDH1 gene), claudin, or occludin. Moreover, there is also increased expression of proteins called “interstitial markers”, such as vimentin or N-cadherin [63]. EMT is a crucial factor in promoting cancer cell progression [64]. It has been reported that low EMT expression blocks GC metastasis through epigenetic modifications. Fujii showed that overexpression of EZH2 caused silencing of the E-cadherin gene in MKN1 cells, while knockdown of EZH2 reversed E-cadherin deletion and downregulated the invasive ability of GC cells [65]. In addition, it has been found that non-coding RNAs (ncRNA) are upstream molecules that regulate EZH2 and can affect downstream signaling pathways that mediate EMT events. Liu showed that HOTAIR recruits and binds PRC2 to inhibit miR34a by epigenetic inheritance, which controls the targets C-Met (HGF/C-Met/Snail pathway) and Snail, contributing to the EMT process in GC cells and accelerating tumor metastasis [66]. TP73-AS1 plays an oncogenic role in Epstein-Barr virus-associated GC (EBVaGC). Using RIP analysis, the authors found that EZH2 directly targets TP73-AS1, suggesting that TP73-AS2 may interact with EZH2 to silence WIFI in an epigenetic manner and trigger EMT events. Furthermore, TP73-AS1 knockdown suppressed EZH2 binding and H3K27me3 levels in the WIF1 promoter, and WIF1 transcription was enhanced [67]. Interestingly, overexpression of EZH2 could reverse the up-regulation of WIF1 mRNA and protein levels induced by TP73-AS1 knockdown, thereby mediating EBVaGC progression. Carvalho et al. [68] revealed the relationship between the miR-101-EZH2 pathway and EMT. The data show that miR-101-downregulated GC cases displayed concomitant EZH2 overexpression (at the RNA and protein levels), which in turn correlates with E-cadherin deletion/abnormal expression. In vitro experiments showed that after transient depletion of EZH2 in KatoII cells using RNAi (Kato-siEZH2), EZH2 transcript levels were reduced, resulting in increased CDH1 mRNA levels. E-cadherin was restored to the plasma membrane of EZH2-deficient cells compared to non-silenced siRNA cells. This strongly suggests that the histone methyltransferase EZH2 mediates the dysfunction of E-cadherin in GC. In addition, EZH2 directly binds to key tumor oncogenes and triggers signaling pathways for EMT events. EZH2 binds to the PTEN locus and downregulates PTEN expression, which activates the Akt pathway, stabilizes vimentin, downregulates E-cadherin, and protects Sox2 and Oct4 from degradation. Thus, this ultimately leads to the acquisition of EMT and pluripotent phenotype in GC cells [54].

EMT-activated transcription factors (EMT-ATF), such as the Snail, Twist, and E-box binding zinc finger protein (ZEB) families, are significant regulators of EMT. The ncRNAs are dysregulated in GC, which can cause EMT events by regulating EMT-ATF. For example, lncRNA CCAT2 downregulates E-calmodulin expression and upregulates ZEB2 expression, which promotes EMT in GC cells. In addition, CCAT2 interacts with EZH2 to regulate the expression of E-cadherin and large tumor suppressor homolog 2 (LATS2) [69,70]. In addition, EZH2 interacts with EMT-ATF to form a multimolecular complex that contributes to the silencing of E-cadherin. Moreover, EZH2 is required to help to recruit Snail-Ring1A/B to E-cadherin promoter sites [71]. The most striking feature of cells that undergo EMT is enhanced cell motility, thus promoting tumor cell invasion and metastasis. In addition, EMT mediates immune evasion and drug resistance in tumor cells. This shows that EMT provides various benefits for tumor growth [72,73].

Another critical event is that EZH2 suppresses tumor growth suppressor genes. Several growth-suppressing genes, including p21, p16, and p27, and pro-death genes, including F-box protein 32 protein (FBOX32), are downstream targets of EZH2 and participate in proliferation, cell cycle arrest, senescence, and apoptosis of tumor cells, and ultimately determines the cell fate [74,75]. Currently, the most studied classical pathway involved in cell cycle alteration is the p53/p21 signaling pathway [76]. As a member of the Cip/Kip family of cyclin kinase inhibitors (CKIs), p21 is a major effector of various tumor inhibition pathways with anti-proliferative activity. It mainly binds and inhibits cyclin-dependent kinases (CDKs) to regulate their biological activity, leading to growth arrest at specific stages of the cell cycle. p21, a key downstream regulator of EZH2, is significantly increased in GC cells with the knockdown of EZH2, resulting in the inhibition of proliferation and invasion of GC cells. In contrast, in GC cells without knockdown of EZH2, EZH2 binds directly to the p21 promoter region. It mediates H3K27me3 modifications that mediate transcriptional repression of P21. Thus, this suggests that EZH2 acts as an oncogene in GC cells by regulating p21 [55]. CXXC finger protein 4 (CXXC4) is a newly discovered GC suppressor and was identified as a new target of EZH2. EZH2 promotes the activation of Wnt signaling by downregulating the expression of CXXC4. The induced aberrant activation of typical Wnt/β-catenin signaling is one of the drivers of progression in many cancers, including GC [77]. Moreover, other tumor suppressor genes that have been studied in GC are targeted by EZH2 for inhibition. For example, EZH2 targeting inhibits CDH1, and the invasion of GC cells is enhanced [65,78]. However, the administration of exogenous CDH1 prevented invasion. In addition, RUNX3 controls the proliferation of gastric epithelial cells. After the knockdown of EZH2, RUNX3 expression was upregulated, and GC cell proliferation was inhibited [78]. These studies suggest that EZH2 can promote the development of gastric carcinogenesis by downregulating the expression of downstream tumor suppressor genes. However, their specific mechanisms of action are unclear and need to be uncovered by further studies.

There is another way of inhibiting cell proliferation by inducing cellular senescence. During senescence, EZH2 is explicitly downregulated in senescent cells, and the deletion of EZH2 has an impact on histone methylation patterns. Deletion of the INK4/ARF gene on chromosome 9p21 is one of the most common cytogenetic events in human cancers [79]. INK4/ARF encodes p15^INK4b^, p14^ARF^, and p16^INK4a^, which are known as common key reprogramming regulators and are inducers of cellular progression toward senescence [80]. Specifically, EZH2 silencing resulted in the loss of H3K27me3 and activation of INK4/ARF genes to some extent, which led to upregulated expression of p15^INK4b^, p14^ARF^, and p16^INK4a^, causing cell cycle arrest and inducing senescence in GC cells [57]. These results clarified that GC cells could escape senescence by recruiting EZH2 to the INK4/ARF locus. Similarly, Bai found that EZH2 suppressed the senescent state in the human GC cells SGC-7901. There is a recovery of phenotypic features of cellular senescence when EZH2 is depleted in cells. Moreover, p21 and p16 were activated to some extent upon EZH2 depletion [55]. EZH2 knockdown causing cellular senescence is an exciting topic. Ito et al. [56] further delved into the mechanism of cellular senescence induced by EZH2 disruption in two broad phases: first, depletion of EZH2 in proliferating cells rapidly initiates a DNA damage response, but significant changes in H3K27me3 do not accompany this phase. Second, concomitant with the eventual deletion of H3K27me3 at a later stage, it induces p16 (CDKN2A) gene expression and effectively activates the senescence-associated secretory phenotype genes (SASP). Here, the gradual depletion of the H3K27me3 marker can be seen as a molecular “timer” that provides a window for cellular repair of DNA damage. Cellular senescence plays an essential physiological role in tumor suppression. Blocking tumor cell cycle progression is one of the important ways to fight against tumors. 

In general, EZH2 plays an active role in EMT events and inhibition of cellular senescence. Moreover, these two significant events facilitate cell invasion and stable growth [81]. EZH2 is an active participant in the occurrence of these two major events that cause the progression of GC. Therefore, given the important role of EZH2 in GC, targeted inhibition of EZH2 expression is a promising measure for treating GC.

## 11. EZH2 Mediates Resistance to Chemotherapeutic Drugs of GC

Chemotherapy is an important measure for treating advanced GC [82]. Due to the emergence of insensitivity and multi-drug resistance (MDR), chemotherapy is not effective in the treatment of GC patients. First-line chemotherapy drugs commonly used clinically for GC, such as oxaliplatin and capecitabine combined chemotherapy, failed in 95% of non-operative patients with GC. Unfortunately, second-line chemotherapeutic agents, such as mitomycin C, irinotecan, adriamycin, methotrexate, or etoposide, also fail to provide better efficacy for patients with GC [83]. Therefore, the drug resistance of GC chemotherapy drugs significantly shortens the survival of GC patients. Thus, there is a great need to understand the mechanisms of GC chemotherapy resistance. The possible mechanisms of drug resistance in GC include reduced drug uptake by GC cells or increased drug efflux; a reduced proportion of active agents in tumor cells due to a reduction in pro-drug activation or an enhancement in drug inactivation; expression and functional changes of molecular targets of anticancer drugs; change in the ability of cancer cells to repair DNA damage induced by anticancer drugs; and expression/function of pro-apoptotic factors or up-regulation of anti-apoptotic genes [83]. 

There are few studies on the EZH2 gene and chemotherapy sensitivity of GC. Some scholars have found that EZH2 is significantly up-regulated in drug-resistant GC cell lines and is involved in regulating the sensitivity of GC to chemotherapy drugs, which is the most widely studied among platinum-based drugs. Zhou investigated the effect of EHZ2 on cisplatin resistance in AGS/DDP cells. He found that EZH2 expression levels were significantly higher in AGS/DDP cells than in the parental cells [84]. In addition, the silencing of EZH2 using siRNA increased the intracellular concentration of cisplatin in AGS/DDP cell lines, which significantly reversed the resistance to cisplatin in AGS/DDP cell lines [84]. In another study, Wang et al. used two GC drug-resistant cell lines, namely vincristine (VCR)-resistant cell lines (SGC7901/VCR) and adriamycin (ADR)-resistant cell lines (SGC7901/ADR) and compared them with the parental cell line SGC7901; miR-126 expression was shown to be decreased in these two resistant cell lines. In contrast, the upregulation of miR-126 expression increased the sensitivity of SGC7901/VCR and SGC7901/ADR cells to VCR and ADR. Mechanistically, the enhancer of EZH2 was identified as a direct target of miR-126, and the silencing of EZH2 reflected the role of miR-126 in drug resistance. However, restoring the EZH2 gene blocks the inhibitory effect of miR-126 on GC [85]. This suggests that the EZH2 gene indeed plays a crucial role in chemotherapeutic agents for GC. In addition, about 20% of lncRNAs can bind and silence the EZH2 gene to increase drug resistance in GC cells. For example, the lncRNA UCA1 up-regulates the level of EZH2 in GC. The over-expressed EZH2 activates the PI3K/AKT pathway [86], which affects the expression of multiple drug resistance-associated and anti-apoptotic proteins and plays a very important role in chemoresistance [86]. Moreover, the knockdown of the EZH2 gene decreased GC cell proliferation, increasing cisplatin-induced apoptosis and caspase-3 expression, inhibiting UCA1-induced upregulation of PI3K/AKT [87]. The scientists found that the lncRNA PCAT-1 is highly expressed in DDP-resistant tissues and cells in GC, which promotes DDP resistance in GC cells by recruiting EZH2 to epigenetically suppress PTEN expression and regulate the miR-128/ZEB1 axis [88,89]. In conclusion, EZH2 is regulated by multiple lncRNAs and is involved in the resistance of GC cells to platinum-based chemotherapy drugs. A final event shared among the mechanisms of action of many antitumor drugs is the activation of apoptosis [83]. It was reported that the apoptotic factors or anti-apoptotic pathways were inhibited in GC cell lines after the knockdown of EZH2. Thus, silencing of EHZ2 could effectively reverse chemotherapeutic drug resistance in GC cells. Based on the above results, targeted silencing of EZH2 can effectively reverse the resistance of GC cells to chemotherapy drugs.

## 12. High Expression of EZH2 Leads to Poor Prognosis for GC Patients

There are many factors affecting the prognosis of GC patients, such as the size of the mass, the depth of tumor infiltration, the pathological type of the tumor, the degree of differentiation of tumor cells, the degree of choroidal invasion, and the metastasis of lymph nodes. It has been reported that EZH2 expression is closely related to the above prognostic factors of GC [6,54,90]. Specifically, EZH2 expression was increased in GC tissues, and the higher the expression level, the higher the malignant degree of the tumor and the worse the prognosis. In a study of patients with GC, the tissue samples of 105 patients with primary GC were included for immunohistochemical detection. Among them, 72 patients had a positive expression of EZH2 protein in GC tissues that was higher than that in para-carcinoma tissue [54]. In addition, the overexpression of EZH2 mRNA and protein detected using qRT-PCR and immunohistochemistry was closely associated with tumor size, lymphatic invasion, and TNM stage. These evidences strongly demonstrated that EZH2 is closely related to the prognosis of GC patients. The Kaplan–Meier method was used to analyze the expression of EZH2 and H3K27me3 proteins and their correlation with the prognosis of GC patients. The results showed that these two highly expressed proteins were commonly found in patients with advanced GC and those with lymph node metastases. Moreover, the overall survival rate of patients with high expression was significantly lower than that of those with low expression. These results are similar to those of other authors [90,91]. Meanwhile, in another more convincing meta-analysis, 872 GC patients were included in this study. The results showed that the expression level of EZH2 protein in GC was higher than that in normal gastric tissue, and positively correlated with tumor-node-metastasis (TNM) stage and lymph node metastasis. The overall survival rate of patients with positive EZH2 expression was shorter than that of patients with negative EZH2 expression [92]. Pan found that high expression of EZH2 in GC tissues was regulated by IL-6/STAT3 signaling. STAT3 acted as a transcription factor to enhance the transcriptional activity of EZH2 by binding to the relevant promoter region (−214~−206), and there is a positive functional loop between STAT3 and EZH2 [91]. Regarding prognosis, STAT3 was positively correlated with EZH2 expression in GC cells and tissues, and activation of EZH2 and STAT3 was significantly associated with the TNM stage and low patient survival. Furthermore, the combination of siSTAT3 and the EZH2-specific inhibitor 3-deazaneplanocin A (DZNep) increased the apoptosis rate of GC cells, suggesting that the combination of siSTAT3 and EZH2 inhibitors may contribute to the potential epigenetic treatment of GC patients. In general, these evidences strongly confirmed the view that high expression of EZH2 may be involved in gastric carcinogenesis. Therefore, the EZH2 protein may be a valuable biomarker for the diagnosis and prognosis of GC. It is meaningful that targeted inhibition of EZH2 expression could contribute to the potential epigenetic therapy against GC patients. 

## 13. Current Status of EZH2 Inhibitor Research

Given that abnormal expression of EZH2 plays an essential role in tumor cell proliferation, invasion, metastasis, and drug resistance [93], targeted inhibition of EZH2/PRC2 is considered an attractive target for cancer therapy. Here, we review some typical EZH2 inhibitors and their current application status. 

DZNep, an inhibitor of S-adenosyl-l-homocysteine (SAH) hydrolase, induces the accumulation of SAH in cells, directly inhibiting histone methyltransferase activity and then indirectly degrading the PRC2 complex. It was one of the first small molecules to evaluate the inhibitory effect of EZH2 [94]. Some studies have shown that DZNep exhibits good anti-tumor properties by inhibiting EZH2 in breast cancer [95], lung cancer [96], prostate cancer [97], colon cancer [98], and other cancer cells. However, DZNep lacks specificity in human tissues because it affects all SAM-dependent processes by blocking the methionine cycle and the regeneration of SAM [99].

In this context, several highly selective EZH2 inhibitors were developed (EPZ005687, GSK126, EPZ6438), which almost always have a 2-pyridone group in their structure [100]. This is mainly because the 2-pyridone group is essential for enzyme inhibition, occupying the site of the common substrate (SAM or the by-product SAH) in the binding pocket of the enzyme [100]. The first highly selective inhibitor was EPZ005687, that competes with SAM and does not compete with peptide substrates, and thereby does not disrupt protein–protein interactions between PRC2 subunits. Most importantly, EPZ005687 was 500 times more selective to the PRC2/EZH2 complex than the other 15 methyltransferases and approximately 50 times more selective to PRC2/EZH2 than PRC2/EZH1 [101]. In lymphomas, EPZ005687 potently reduced H3K27me3 levels in EZH2 mutation-containing cells [94].

Another highly selective inhibitor is GSK126, which has a similar core structure to EPZ005687, but GSK126 is more than 1000 times more selective for EZH2 than the other 20 human methyltransferases [101]. GSK126 effectively suppresses the expression of H3K27me3 in cells with EZH2 mutations. McCabe et al. showed that GSK126 effectively inhibited the proliferation of EZH2 mutant diffuse large B cell lymphoma (DLBCL) cell lines and markedly inhibits the growth of EZH2 mutant DLBCL xenografts in mice. In addition, tumor growth was essentially completely inhibited (91–100% inhibition) in subcutaneous xenografts utilizing the more aggressive KARPAS-422 and Pfeiffer cells when high doses of GSK126 were administered (300 mg/kg twice a week) [102]. In 2014, GSK126 entered phase I clinical trials in patients with various lymphomas and solid tumors [101]. In 2019, the results of the phase 1 clinical study of GSK126 (NCT02082977) were published [103]. However, the trial was forced to stop due to the deficiency of clinical activity of the drug. This is mainly because of the limitation of twice-weekly administration and the pharmacokinetic characteristics of the drug itself. Nevertheless, GSK126 remains a promising agent. There is a mistaken perception that GSK126 may inhibit tumor immunity due to GSK126 treatment essentially inhibiting tumor growth in immunodeficient hosts but not in immunocompetent hosts. In another study, however, GSK126 exerted anti-tumor activity in immunologically active hosts combined with drugs (gemcitabine and 5-fluorouracil) that deplete myeloid-derived suppressor cells [104]. These results provide a new strategy for the use of GSK126 in the clinic. 

In 2013, UNC1999 was reported to be the first oral bio-availability inhibitor with high in vitro potency. UNC1999 showed high efficacy in vitro against wild-type and mutant EZH2 and EZH1, effectively reducing the level of H3K27me3 in cells and selectively killing diffuse large B-cell lymphoma cell lines containing EZH2^Y641N^ mutation with low cytotoxicity [105]. Currently, UNC1999 is used to study mixed lineage leukemia [106].

As a result of this effort, Tazemetostat, an oral, first-in-class inhibitor of EZH2, was introduced to strongly inhibit wild-type and mutant EZH2 enzyme activity with its improved potency, pharmacokinetic parameters, and oral bioavailability [107]. Tazemetostat is a direct inhibitor of EZH2, which can competitively bind to the SET domain of EZH2 protein directly with SAM. Tazemetostat has mainly been used in lymphoma and epithelioid sarcoma (ES) studies [108]. The results of a phase I clinical study conducted in France showed that tazemetostat exhibited significant antitumor activity in patients with non-Hodgkin lymphoma (NHL) and advanced solid tumors [109]. The recommended dose of tazemetostat was 800 mg, administered twice a day, and is more effective in NHL. Ribrag et al. [110] suggested that tazemetostat showed apparent clinical efficacy in DLBCL, follicular lymphoma (FL), and marginal zone lymphoma (MZL) patients, showing good anti-tumor activity, with the best effect in patients with stable FL. In a study of tazemetostat for the treatment of epithelioid sarcoma (ES), phase II clinical data showed an overall efficacy rate of 15% and a disease control rate of 26% with tazemetostat administration (2.4 to 18.4 months). Further analysis of clinical study data presented by Epizyme at the 2019 American Society of Clinical Oncology showed that tazemetostat improved life expectancy in ES patients and those first-treatment ES patients taking tazemetostat had better outcomes than patients with relapsed or refractory ES [111]. These results suggest that tazemetostat shows significant anti-ES activity and may be a new treatment option for ES patients. Ultimately, tazemetostat received accelerated approval in January 2020 in the USA for treating adults and adolescents aged ≥16 years with locally advanced or metastatic ES not eligible for complete resection [111]. 

The polyprotein nature of the PRC2 complex and EZH2 function is significantly dependent on those of other core subunits, such as EED. Therefore, scientists developed a compound that binds to EED. On the one hand, the possibility of weakening PRC2 complex function by interfering with the close protein-protein interaction (PPI) between EZH2 and EED has facilitated the development of different chemical types as inhibitors of EZH2−EED. These chemical agents exert a methyltransferase inhibitory activity on PRC2 by impeding the scaffolding role of EED on the SET domain of EZH2 [112]. High-throughput screening demonstrated the “druggability” of the H3K27me3 recognition cavity of EED as a means of heterologous inhibition of EZH2 catalysis. Eventually, the Novartis company developed an EED-binding agent called MAK683 [113], which is currently in phase I/II study (NCT02900651) for the treatment of patients with DLBCL, nasopharyngeal carcinoma, or other advanced solid tumors of malignancy [114]. Some of the information on these inhibitors is summarized in Table 4.

In conclusion, some inhibitors targeting EZH2 have achieved some success, but most anti-tumor studies of EZH2 inhibitors are still in the preliminary stages. With the continuous deepening of the research and the gradual extension to the clinic, the molecular mechanism of the anti-tumor effect of EZH2 small molecule inhibitors will be further clarified. In the future, it is necessary to develop efficient, highly selective, and low-toxicity EZH2 inhibitors, which are an important target for cancer therapy. 

## 14. Conclusions

EZH2 is highly expressed in GC, which has been proven to be associated with poor prognosis in GC. These epigenetic disorders are frequently mutated by multiple factors in GC and other cancers, resulting in the uncontrolled expression of many downstream cancer-associated genes. Therefore, it is meaningful that targeting these epigenetic regulators may have positive implications for treating some tumors. Many drugs targeting EZH2/PRC2 are being developed and evaluated in clinical trials. However, most are still in preclinical studies or phase 1/2 clinical trials, with only tazemetostat approved for the treatment of epithelioid sarcoma (ES) and preliminary evidence of efficacy in follicular lymphoma (FL). Although the role of EZH2 in GC has achieved positive results, most of the research has studied the pre-clinical stage of targeted treatment of EZH2, which has not yet broken through to the clinical stage. This is mainly because many of the modification enzymes of EZH2 and the exact sites of PTMs are unknown. In addition, it is not known whether rare types of PTMs exist in EZH2 such as succinylation, malonylation, crotonylation, propionylation, and butyrylation. More importantly, there are no clinical trials targeting EZH2 PTMs for cancer treatment. At present, targeted therapies for EZH2 are mostly focused on the hematologic and lymphatic systems, such as B-cell lymphoma and non-Hodgkin’s lymphoma. We may be inspired by studies that have achieved some results, such as combining EZH2 inhibitors with immunotherapy, chemotherapy, targeted therapy, endocrine therapy, and other therapies that may achieve complementary or synergistic anti-tumor effects [115]. In conclusion, as a novel target for GC treatment, EZH2 has become a research hotspot, and its functions and effects have been continuously revealed. In the future, it is necessary to further study its mechanism of action and develop therapeutic drugs based on this target.

## Figures and Tables

**Figure 1 cancers-15-00425-f001:**
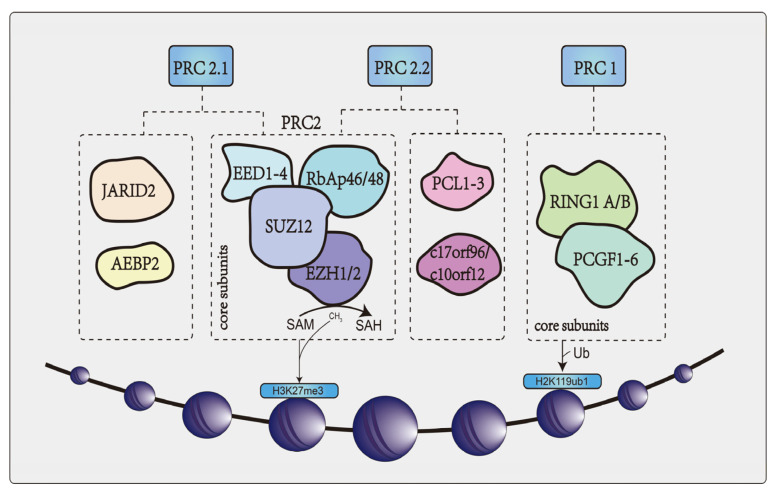
The core structure of PRCs, including PRC1 and PRC2. PRC1 complexes are E3 ubiquitin ligases that monoubiquitinate lysine 119 of histone H2A (H2AK119ub1), consisting mainly of two core subunits. PRC2 consists of four major core subunits and binds to different non-core subunits divided into PRC2.1 and PRC2.2. PRC2 catalyzes the monomethylation, dimethylation, and trimethylation of lysine 27 on histone H3 (H3K27me1, H3K27me2, and H3K27me3). SAM provides the methyl group for the reaction catalyzed by histone methyltransferase.

**Figure 2 cancers-15-00425-f002:**
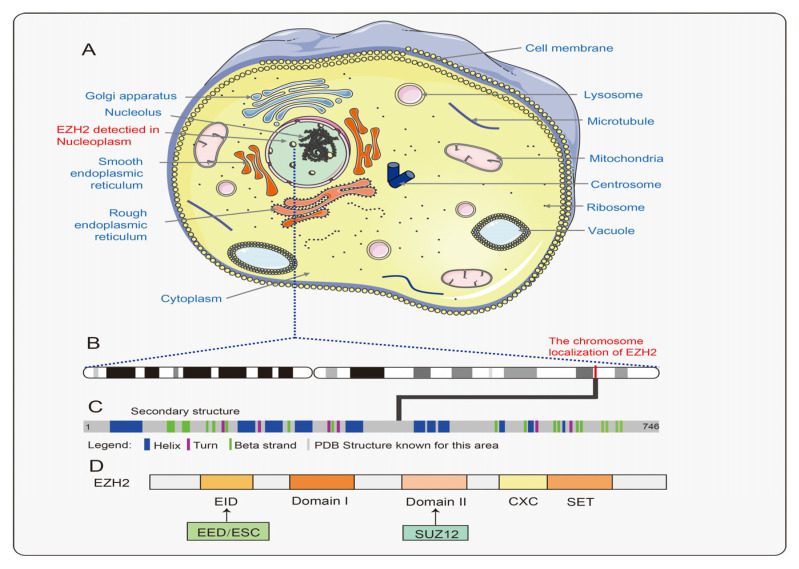
The gene and protein structure of the EZH2. (**A**) The location of the EZH2 protein in the cell. (**B**) The chromosome localization of EZH2. (**C**) The protein secondary structure of EZH2. (**D**) Schematic representation of the organization of the five functional domains in EZH2 is depicted. The EID structural domain is the binding site for the EED subunit in the RC2 complex. The domain II structural domain is the linkage site for the SUZ12 subunit in the PRC2 complex. The SET structural domain is the site that exerts methyl transfer activity and is also the binding site for SAM. The CXC structural domain also contributes to methyl transfer activity, whereas the function of the domain I structural domain is not known.

**Figure 3 cancers-15-00425-f003:**
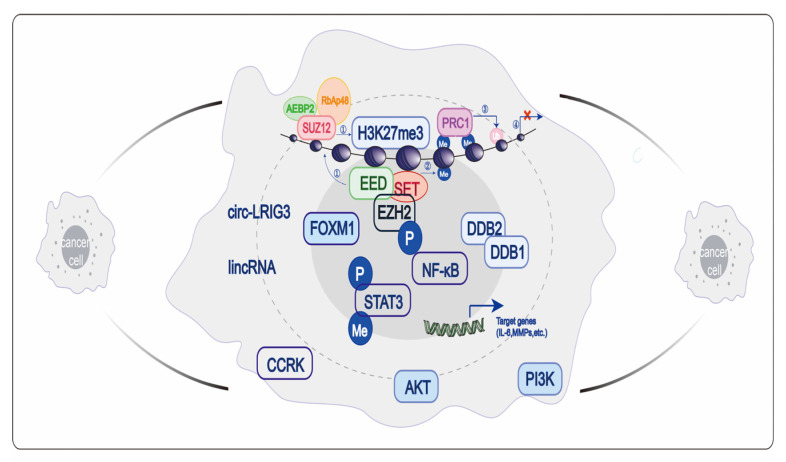
Mechanism of abnormal expression of EZH2 in cancer. The figure mainly shows the regulation of EZH2 at the level of transcription and translation. DDB: damage specific DNA binding protein; CCRK: cell cycle-related kinas; FOXM1: forkhead box protein M1.

**Table 1 cancers-15-00425-t001:** The genes that are regulated by EZH2 through histone modifications.

Genes	Mechanism of Action of EZH2	The Role of Genes	Reference
*METTL3*	EZH2 overexpression leads to increases in H3K27me3, up-regulating the expression of *METTL3*	Drug resistance	[25]
*P16*	EZH2 overexpression leads to increases in H3K27me3, inhibiting the expression of *P16*	Inhibition of tumor growth; Drive cellular differentiation	[26]
*E-cadherin*	EZH2 overexpression leads to increases in H3K27me3, inhibiting *E-cadherin*	Inhibition of tumor growth	[26]
*HIF-1α*	EZH2 stabilizes the expression of *HIF-1α*	Promotion of tumor growth and metabolism favoring glycolysis	[5]
*INK4B-ARF-INK4A*	EZH2 suppresses the expression of *INK4B–ARF–INK4A*	Induce cell cycle progression and inhibit cell senescence	[27]

Note: *METTL3*: methyltransferase-like 3; *HIF-1α*: hypoxia inducible factor-1.

**Table 2 cancers-15-00425-t002:** The genes that are regulated by EZH2 through DNA methylation.

Genes	Mechanism of Action of EZH2	The Role of Genes	Reference
*BMPR1B*	EZH2 represses the expression of BMPR1B	Inhibition of growth of tumors	[30]
*VASH1*	EZH2 represses VASH1	Promotion of the angiogenesis of tumors	[31]
*SRBC*	EZH2 plays a substantial role in silencing SRBC	Inhibition of tumor growth; involved in tumor resistance against chemotherapeutic agents	[32]
*RASSF5*	EZH2 inhibits RASSF5	Suppression of cell growth	[33]
*ITGB2*	EZH2 inhibits ITGB2	Contribute to natural killer cell development and function	[34]

Note: *BMPR1B*: bone morphogenetic protein receptor type-1B; *VASH1*: vasohibin1; *SRBC*: serum deprivation response factor-related gene product that binds to the c-kinase; *RASSF5*: ras association domain family member 5; *ITGB2*: integrin beta 2.

**Table 3 cancers-15-00425-t003:** The genes related to EZH2 and gastric carcinogenesis.

Genes	Mechanism of Action of EZH2	The Role of Genes	Reference
*E-cadherin*	EZH2 causes the silencing of the E-cadherin gene	E-cadherin is involved in epithelial–mesenchymal transition, causing GC metastasis	[53]
*PTEN*	EZH2 downregulates PTEN expression	PTEN causes GC metastasis	[54]
*P21*	p21 increases when EZH2 is knocked down	p21 inhibits proliferation and invasion of GC cells	[55]
*P16*	p16 increases when EZH2 is knocked down	p16 promotes GC cellular senescence	[56]
*INK4/ARF*	EZH2 silencing results in the activation of INK4/ARF	INK4/ARF causes cell cycle arrest and induces senescence in GC cells	[57]

*PTEN*: Phosphatase and tensin homolog deleted on chromosome ten.

**Table 4 cancers-15-00425-t004:** Pre-clinical and clinical trial status of drugs related to EZH2.

Drug	Role	Phase	Reference(s)
DZNep	SAH hydrolase inhibitor	pre-clinical	[94,99]
EPZ005687	Inhibitor of EZH2 T641 and A677 mutants	pre-clinical	[101]
GSK126 (GSK2816126)	SAM-competitive inhibitors of EZH2	Phase I	[102]
Tazemetostat (EPZ-6438, E7438)	SAM-competitive inhibitors of EZH2	Phase I/II	[107]
UNC1999	SAM-competitive inhibitors of EZH2 and EZH1	pre-clinical	[105]
MAK683 (EED226)	Selective EED inhibitor	Phase I/II	[113]

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
