# Peer review of "EZH2: An Accomplice of Gastric Cancer"

_cancers, 2023, doi:10.3390/cancers15020425_

Round 1
Reviewer 1 Report
In the manuscript: “ EZH2: an accomplice of gastric cancer”, the authors want to provide a broad overview of all the information inherent in the correlation between EZH2 and gastric cancer.
The work is certainly interesting but not clearly described; it needs more revisions so that the reader can quickly get the information the authors want.
So in this regard, I invite the following corrections and implementations of the manuscript.
- In the section Introduction (lane 30), you can start with GC the sentence because you have already described it in the abstract abbreviation.
- In section 2. Overview of EZH2 (lane 81) I suggest inserting a schematic and representative figure containing the gene and protein structure of the EZH2 to be included.
- In section 3. Histone modification of EZH2, It would be necessary to indicate in a Table which genes are regulated by EZH2 through histone modifications.
- In section 4. DNA methylation of EZH2, they need an additional Table to describe genes regulated by EZH2 through DNA methylation and insert them.
- Why are the legends inserted 2 times? One in the image and one in the text? One in the text is sufficient.
- At lane 229, Francesco is the name of n. 50 of reference author, not the surname.
- In section 9. The authors cite only three references in the association of the EZH2 gene with genetic susceptibility to GC, which seem to me to be few.
- In section 10. Correlation of EZH2 gene with gastric carcinogenesis and invasive metastasis, a Table showing genes related to EZH2 and gastric carcinogenesis should be included.
Author Response
Dear Reviewer:
Thank you very much for your careful review of the manuscript. Your suggestions are very meaningful to me. I will revise the article one by one according to your suggestions. Next, I will elaborate on the specific changes made to the article.
- In the section Introduction (lane 30), you can start with GC the sentence because you have already described it in the abstract abbreviation.
Thank you for your reminding. We have referred to the submission guide again which suggested that we use less abbreviations in the abstract. So we did not use abbreviations in the abstract. Finally, the word “Gastric cancer (GC)” for the text appears for the first time and is abbreviated, so we don't change it.
- In section 2. Overview of EZH2 (lane 81) I suggest inserting a schematic and representative figure containing the gene and protein structure of the EZH2 to be included.
The picture has been drawn and inserted into the text.
- In section 3. Histone modification of EZH2, It would be necessary to indicate in a Table which genes are regulated by EZH2 through histone modifications.
The table has been summarized and inserted into the text.
- In section 4. DNA methylation of EZH2, they need an additional Table to describe genes regulated by EZH2 through DNA methylation and insert them.
The table has been summarized and inserted into the text.
- Why are the legends inserted 2 times? One in the image and one in the text? One in the text is sufficient.
The extra legends have been removed and there is now only one in the text.
- At lane 229, Francesco is the name of n. 50 of reference author, not the surname.
The author's surname has been amended.
- In section 9. The authors cite only three references in the association of the EZH2 gene with genetic susceptibility to GC, which seem to me to be few.
As for this part, we have tried to find other literatures on this aspect, but generally speaking, the literatures are limited, so we have added a few contents.
- In section 10. Correlation of EZH2 gene with gastric carcinogenesis and invasive metastasis, a Table showing genes related to EZH2 and gastric carcinogenesis should be included.
The table has been summarized and inserted into the text.
Thank you again for your review. The above descriptions were all my revisions to the manuscript. This manuscript means a lot to me and we are highly expect to that the manuscript is accepted for publication in the Journal.
We look forward to hearing from your further decision. Thank you!
Kind regards,
Youcheng Zhang

Reviewer 2 Report
In this paper the Authors introduce EZH2 and clarify the mechanism of abnormal expression of EZH2 in cancer. They also summarize the role of EZH2 in GC onset, metastasis, evolution and explain how EZH2 mediates the resistance to chemotherapeutic drugs . Last but not least, clarify the state of art of anti EZH2 targeted therapy.
In this way the Authors cover all the fields of CG develompent and clearly explain the effect of EZH2 deregulation as well as the therapeutic opportunity correlated to EZH2 inhibition
Author Response
Dear Reviewer:
Thank you very much for your careful review of the manuscript. We further modified some sentences in the manuscript to make it fluent. Some tables and a picture have also been added which can give readers a clearer understanding of EZH2's role in gastric cancer.
Thank you again for your review. The above descriptions were all my revisions to the manuscript.
We look forward to hearing from your further decision. Thank you!
Kind regards,
Youcheng Zhang

Reviewer 3 Report
In this review manuscript, the authors summarized the role of EZH2 in gastric cancer. EZH2 is the cata-14 lytic subunit of the polycomb repressor complex 2 (PRC2), which trimethyles histone 3 at Lys-27 15 and regulates the expression of downstream target genes through epigenetic mechanisms. In gastric cancer, EZH2 is highly expressed to promote cancer progression through multiple pathways. Thus, targeting EZH2 might be a potential therapeutic strategy to inhibit cancer development. In this review, the authors summarized the role of EZH2 in gastric cancer and the clinical trials targeting EZH2 activities. As too many reviews summarized the role of EZH2 in cancers, and pointed to the role of EZH2 in the multiple hallmarks, including metabolism and immune escape. Even the strategies for targeting EZH2 have been comprehensively discussed from the protein structure, and from the combinational therapy. It is hard to find a novel angle in this review, except the authors link the distinct molecular features of gastric cancer with EZH2, such as h.pylori infection, EBV infection, intestinal- diffuse- gastric cancer, MSI subtypes, etc…
Author Response
Dear Reviewer:
Thank you very much for your careful review of the manuscript. As you said, many reviews have been published on EZH2, but we found no reviews in gastric cancer. We described the effects of EZH2 on gastric cancer from multiple perspectives, aiming to provide research ideas for the study of gastric cancer epigenetics. Some tables and a picture have also been added which can give readers a clearer understanding of EZH2's role in gastric cancer. In addition, We optimized the description of some statements for better understanding. Besides, what you mentioned needs to be linked to he distinct molecular features of gastric cancer, and we have indeed considered this content, but there are really too few studies related to EZH2. In future research work, we will continue to track this content in the future.
Thank you again for your review. The above descriptions were all my revisions to the manuscript.
We look forward to hearing from your further decision. Thank you!
Kind regards,
Youcheng Zhang

Round 2
Reviewer 1 Report
Dear Authors,
as a result of the changes made, the manuscript presents a clearer organization.
The work can be accepted for publication.
Reviewer 3 Report
No much other comments, but I still wanna see the EZH2 genetic/epigentic information with distinct GC features.